# Research on Diesel Engine Fault Diagnosis Method Based on Stacked Sparse Autoencoder and Support Vector Machine

Huajun Bai [1], Xianbiao Zhan [1,2], Hao Yan [1], Liang Wen [1], Yunbin Yan [1] and Xisheng Jia [1,*]

1   Shijiazhuang Campus, Army Engineering University of PLA, Shijiazhuang 050003, China;
    baihuajun001@gmail.com (H.B.); zhanxianbiao001@gmail.com (X.Z.); yanhao202201@gmail.com (H.Y.);
    wenliang202201@gmail.com (L.W.); yanyunbin001@gmail.com (Y.Y.)
2   Tai'an Intelligent Water Industry Technology Research Institute, Tai'an 271024, China
*   Correspondence: jiaxisheng001@gmail.com; Tel.: +86-0311-87994788

**Abstract:** Due to the relative insufficiencies of conventional time-domain waveform and spectrum analysis in fault diagnosis research, a diesel engine fault diagnosis method based on the Stacked Sparse Autoencoder and the Support Vector Machine is proposed in this study. The method consists of two main steps. The first step is to utilize the Stacked Sparse Autoencoder (SSAE) to reduce the feature dimension of the multi-sensor vibration information; when compared with other dimension reduction methods, this approach can better capture nonlinear features, so as to better cope with dimension reduction. The second step consists of diagnosing faults, implementing the grid search, and K-fold cross-validation to optimize the hyperparameters of the SVM method, which effectively improves the fault classification effect. By conducting a preset failure experiment for the diesel engine, the proposed method achieves an accuracy rate of more than 98%, better engineering application, and promising outcomes.

**Keywords:** diesel engine; fault diagnosis; stacked sparse autoencoder; support vector machine

## 1. Introduction

Diesel engines have been widely utilized in machinery construction, heavy trucks, electric power generation, petrochemical industries, and military equipment. In practice, once the malfunction status of the diesel engine has been obtained, production and maintenance can be better arranged. Not only can the loss of unplanned shutdowns be effectively reduced, but also more effective maintenance can be implemented, based on the status information, further reducing maintenance costs. The fault diagnosis of a diesel engine involves a series of steps, including signal detection, fault-type judgment, fault location, and fault recovery. Fault-type judgment has long been the focus of several types of research. Due to the complexity of the fault signal characteristics of a diesel engine, both the accuracy and the timeliness of fault-type judgments have not been resolved well. The maturity of the technology has not reached the level that industrial applications expect, which limits the application of technologies such as fault prediction and health management (PHM), when the management of diesel engine equipment is under consideration.

In recent years, researchers have conducted much research on the fault diagnosis of diesel engines, and have achieved better outcomes [1–5]. The process of acquiring vibration signals is both simple and convenient, and the fault diagnosis speed is fast, as disassembly is not required for analysis. At the same time, due to the rapid development of sensor technology, feature extraction technology, and failure pattern recognition methods, the accuracy of fault diagnosis has been effectively improved [6]. Therefore, fault diagnosis based on vibration signals has become an effective data-driven analysis method [7,8].

The state signal of the diesel engine, collected by vibration sensors, usually has non-stationarity, nonlinearity, and complexity properties. The data needs to be preprocessed, and fault features need to be extracted. The implementation of dimension reduction

methodology provides an effective method for better fault diagnosis. This technology includes methods such as linear discriminant analysis [9], principal component analysis (PCA) [10], and popular learning methods [11]. Javad, et al. [12], used principal component analysis to reduce the size of the data set, and to eliminate the possible singularity of the data set. Jianbo, et al. [13], proposed a novel manifold learning algorithm, combining both global and local/non-local discriminant analysis methods. Both dimension reduction and comparative analysis of the fault features verified the superiority of the method, for mechanical fault diagnosis. Yuncheng, et al. [14], proposed a fault detection and diagnosis method based on principal component analysis, utilizing the empirical pattern decomposition. The PCA model was employed to reduce the dimension of the historical data, and both the accuracy and effectiveness of the method were verified by experiments. Hongmei, et al. [15], considered that the gearbox fault feature extraction method was based on empirical mode decomposition and multifractal detrend cross-correlation analysis. The PCA model was utilized to decrease the dimension of the extracted multifractal fault feature vectors. The experimental results showed that the method could effectively distinguish different fault modes. When linear methods performing feature extraction on nonlinear data were under consideration, the distribution rules between the data could not be effectively found, yielding poor features results. Therefore, for nonlinear data, this paper proposes the Stacked Sparse Autoencoder (SSAE) feature fusion method, to realize the conversion of nonlinear correlation into linear correlation, so that the feature dimension reduction is achieved better.

After the SSAE method is implemented to fuse the fault features and reduce the dimension, an effective classification method is required, to accurately identify the diesel engine faults. The Support Vector Machine has a solid mathematical foundation, stable calculation property with a high success rate, and can achieve nonlinear separation. This paper therefore adopted the SVM method for classification. C, et al. [16], proposed the MVMD band energy method, utilizing a four-channel vibration signal for fault diagnosis by extracting energy fault eigenvalues and employing a Support Vector Machine to diagnose and identify faults. Zhao, et al. [17], developed a multi-condition fault diagnosis method based on the optimized Mel frequency cepstrum, which combined features with modal decomposition, and employed the nearest neighbor classifier for training and identification. Cai, et al. [18], proposed a new method to diagnose the faults of marine diesel engines. The diesel engine was divided into four subsystems, and the Support Vector Machine algorithm was employed to classify the faults of each subsystem. Kun, et al. [19], proposed a diesel engine method for valve train fault diagnosis, with a variational stacking autoencoder and a harmony search optimizer, and the classification method could effectively identify faults. Zhang, et al. [20], proposed a complete ensemble intrinsic time scale decomposition method, combined with the least squares Support Vector Machine classification method, that was optimized by particle swarm optimization to diagnose and identify faults. Meghdad, et al. [21], proposed a data mining technique and a novel data fusion method utilizing an artificial neural network for fault identification. Zhong K, et al. [22], proposed a local Fisher discriminant analysis based on the sparse kernel, which effectively improved the accuracy of fault diagnosis.

Although fault diagnosis technology has made great progress, it also has some shortcomings, as follows:

1.  The artificial neural network not only needs to rely on a large number of training samples but also has an issue called overfitting or local optimal solution;
2.  Although the Support Vector Machine resolves the problem of small samples, it still has similar shortcomings to those of the artificial neural network.

The related literature has done exploratory research on SVM fault diagnosis, which proves the feasibility of utilizing the SVM method to diagnose faults. However, the influence on the results of the fault diagnosis, of changing the number of sensors installed, their installation positions, and the extraction of vibration features, was previously found to be insufficient. This paper proposes a fault diagnosis method for a diesel engine based

on the SSAE-SVM, to fill this gap. By utilizing the SSAE feature fusion method to reduce the data dimension, the grid search and K-fold cross-validation methods were employed to optimize the hyperparameters of the SVM method, which effectively improved the fault classification effect. Finally, the feasibility and effectiveness of the method are shown through preset fault experiments.

The main contributions of this paper are summarized as follows:

1.  The sensor combination analysis provides a reference for the optimal layout of sensors. In the case of utilizing fewer sensors, higher diagnostic accuracy is obtained, while diagnostic costs are reduced;
2.  By using the characteristic parameter analysis when fewer sensors are adapted, the representative characteristic parameters of diesel engine fault diagnosis can be effectively extracted, and a better diagnosis effect is obtained.

The second part of this paper describes the basic principles of the feature dimension reduction and fault diagnosis; the third part introduces the process of the fault diagnosis method for a diesel engine based on the SSAE-SVM; the fourth part verifies the effectiveness of the proposed method through the preset fault experiment; the fifth part summarizes this research.

## 2. The Basic Principles of the Feature Dimension Reduction and Fault Diagnosis

### 2.1. The Fundamentals of the SSAE

The Autoencoder (AE) is a neural network with a hidden layer of unsupervised feature learning. Its core structure is to utilize one or more layers of neural networks to map the input data to obtain the output vector, as shown in Figure 1. Moreover, the Autoencoder can be utilized to decrease the dimension of data features, which can characterize both linear and nonlinear transformations. The unlabeled input vector is subjected to weighted mapping to obtain the value of the output vector of the hidden layer through the Autoencoder. Its functional representation is expressed by:

$$y_i = f_\theta(x_j) = S(\sum_{j=1}^{N} W_{ij}x_j + b_i) \tag{1}$$

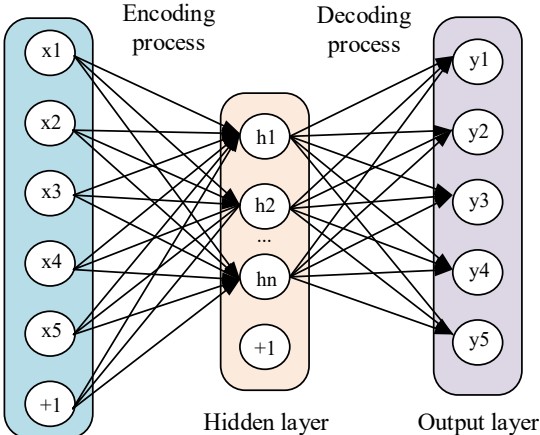

**Figure 1.** Autoencoder: the schematic diagram of both the encoding and the decoding process.

In Equation (1), $y_i$ is the activation value of the hidden layer, $W_{ij}$ is the weight coefficient, $b_i$ is the offset vector of the hidden layer, and $S(x)$ represents the activation function. The sigmoid function is employed herein. The minimized reconstruction error is defined by:

$$L(x_i, y_i) = \frac{1}{2}\|x_i - y_i\|^2 \tag{2}$$

Weight parameters from the input layer to the hidden layer are denoted by $\theta = \{W, b\}$, and Weight parameters from the hidden layer to the output layer are denoted by $\theta' = \{W', b'\}$.

The Sparse Autoencoder (SAE) is an improvement based on the Autoencoder. When there are few neurons in the hidden layer, they are taken as features to achieve data dimension reduction. When there are many neurons in the hidden layer, a sparsity restriction is added, to train the network and extract valuable features. The sparsity limitation is that the processing of the hidden layer neurons is inhibited most of the time. The sparsity restriction introduces a cost function [21], denoted by:

$$J_s = J + \beta \sum_{j=1}^{s_2} K_L(\rho \| \rho_j)c \tag{3}$$

In Equation (3): $\rho$ is the sparsity parameter; $S_2$ represents the number of neurons in the hidden layer; $\beta$ represents the penalty factor for controlling sparsity; and $K_L(\rho \| \rho_j)$ represents the method to measure the difference between $\rho$ and $\hat{\rho}_j$.

Usually, a simple Sparse Autoencoder is not ideal for training, Therefore, this paper adopted a stacking method of training each hidden layer separately with the unsupervised learning of multiple Sparse Autoencoders, and connecting these layers to form a stacked network. The fusion process based on the fault characteristics of the SSAE is depicted in Figure 2. Each sensor extracted 31 parameter characteristics from the collected vibration signals, and represented them as $31 \times N$ input feature vector into the SSAE. After conducting the SSAE feature fusion, 31 new feature vectors were obtained.

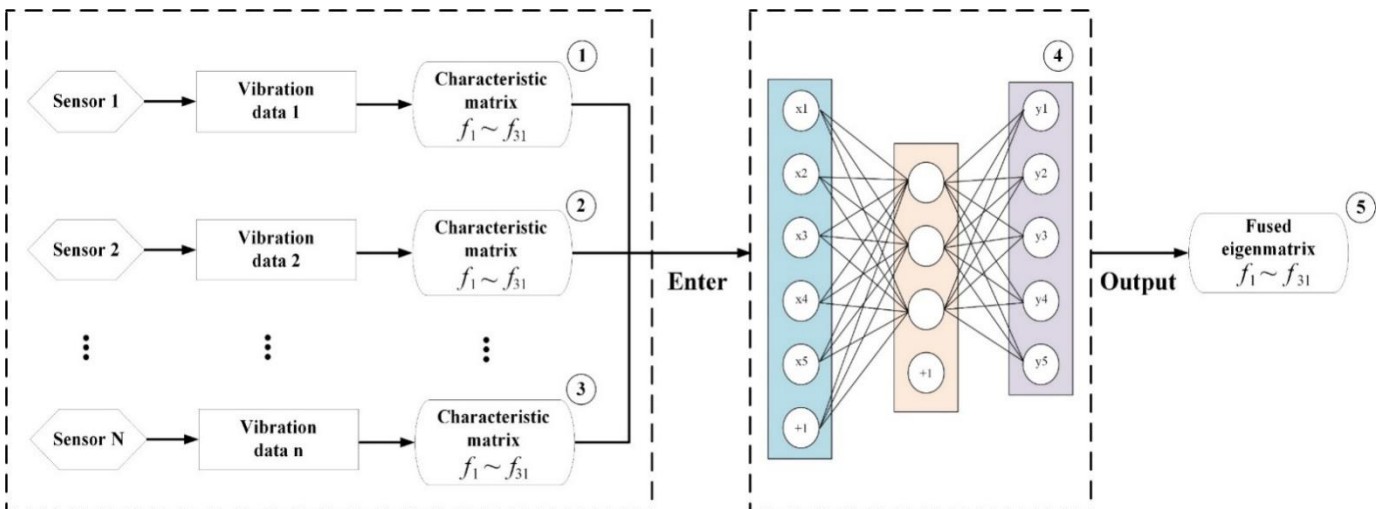

**Figure 2.** The SSAE-based fault feature fusion process.

### 2.2. The Fundamentals of the SVM

The basic principle of the Support Vector Machine is to find an optimal classification hyperplane that can separate the two types of sample data, and maximize the distance between the classified ones. Figure 3 shows these two types of data samples, and $W$ represents the hyperplane separating them. The separated hyperplanes are then moved horizontally to both sides. The critically separated hyperplanes $W_1$ and $W_2$ can effectively divide the samples into two types. The distance between them is called the classification interval. The optimal classification hyperplane can effectively separate these two types of samples, so that the classification interval becomes the largest.

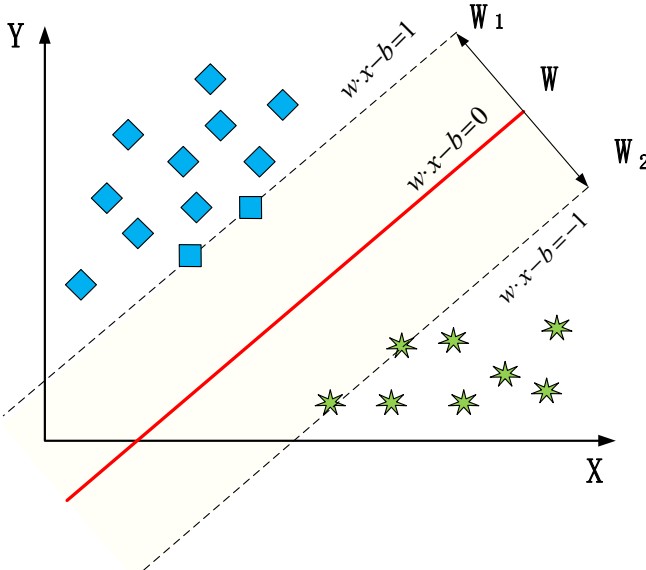

**Figure 3.** The optimal classified hyperplane based on the SVM.

A separating hyperplane equation in the sample is determined in the mathematical form of $w \cdot x + b = 0$. According to the requirement of the maximum separation hyperplane, the problem, in the form of an optimization setting, is expressed by [23]:

$$\begin{cases} \min \frac{1}{2}\|w\|^2 \\ s.t. \ y_i(w \cdot x + b) \geq 1, \ i = 1, 2, \cdots, n \end{cases} \tag{4}$$

The optimal hyperplane to be found in Equation (4) is represented by $w \cdot x + b = 0$: $b$ is the offset vector; w represents the normal vector of the optimal hyperplane. To resolve the optimization problem, the Lagrange multiplier method is employed to find its dual form. Afterward, this problem is transformed into a convex quadratic programming problem with inequality constraints. The specific solution method is referred to in the literature [23].

### 2.3. The Fundamentals of the Grid Search and K-Fold Cross-Validation Optimization

Penalty factor $C$ and kernel parameter $g$ play a very critical role in the classification effect of the SVM. When the parameters that are not optimal are selected, the classification outcome may be unsatisfactory. The value of the kernel parameter $g$ can directly affect the accurate segmentation of the dataset to be classified. The smaller the value of the kernel parameter $g$, the rougher the classification of the data set will be, which may cause the data not to be effectively distinguished. Thus, under-fitting could easily occur. The main function of the penalty factor $C$ is to balance the structural risk and the empirical risk. When the value of the penalty factor $C$ is low, the structural risk is low too, so the corresponding empirical risk is higher, and under-fitting is more likely to occur. Otherwise, over-fitting is prone to occur. Therefore, the combination of grid search and K-fold cross-validation is employed to optimize the penalty factor $C$ and the kernel parameter $g$ concurrently.

Grid search is a method of traversing specific combinations to optimize model performance and K-fold cross-validation (that is, all training samples are divided into K parts, then one part is selected in turn for testing; the remaining K−1 parts are used for training, and the verification is repeated K times in total, usually set K = 10). To effectively prevent the model from over-fitting, the optimal hyperparameters $C$ and $g$ are resolved, shown in Figure 4.

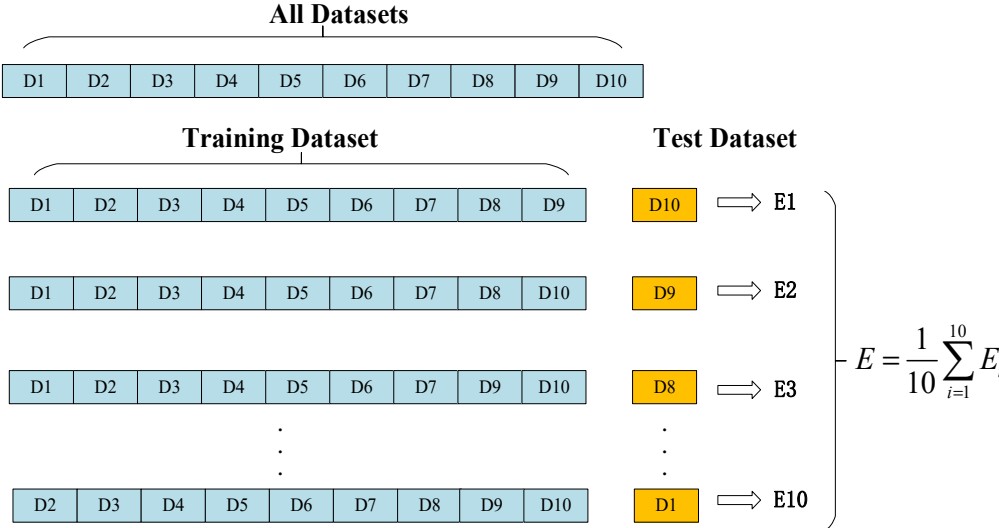

**Figure 4.** Schematic diagram of grid search and K−fold cross−validation.

## 3. The Process Flow of the Fault Diagnosis Method of a Diesel Engine Based on the SSAE-SVM

By combining both SSAE and SVM methods, this research proposes a diesel engine fault diagnosis approach based on the SSAE-SVM. The steps of the method are presented in Figure 5.

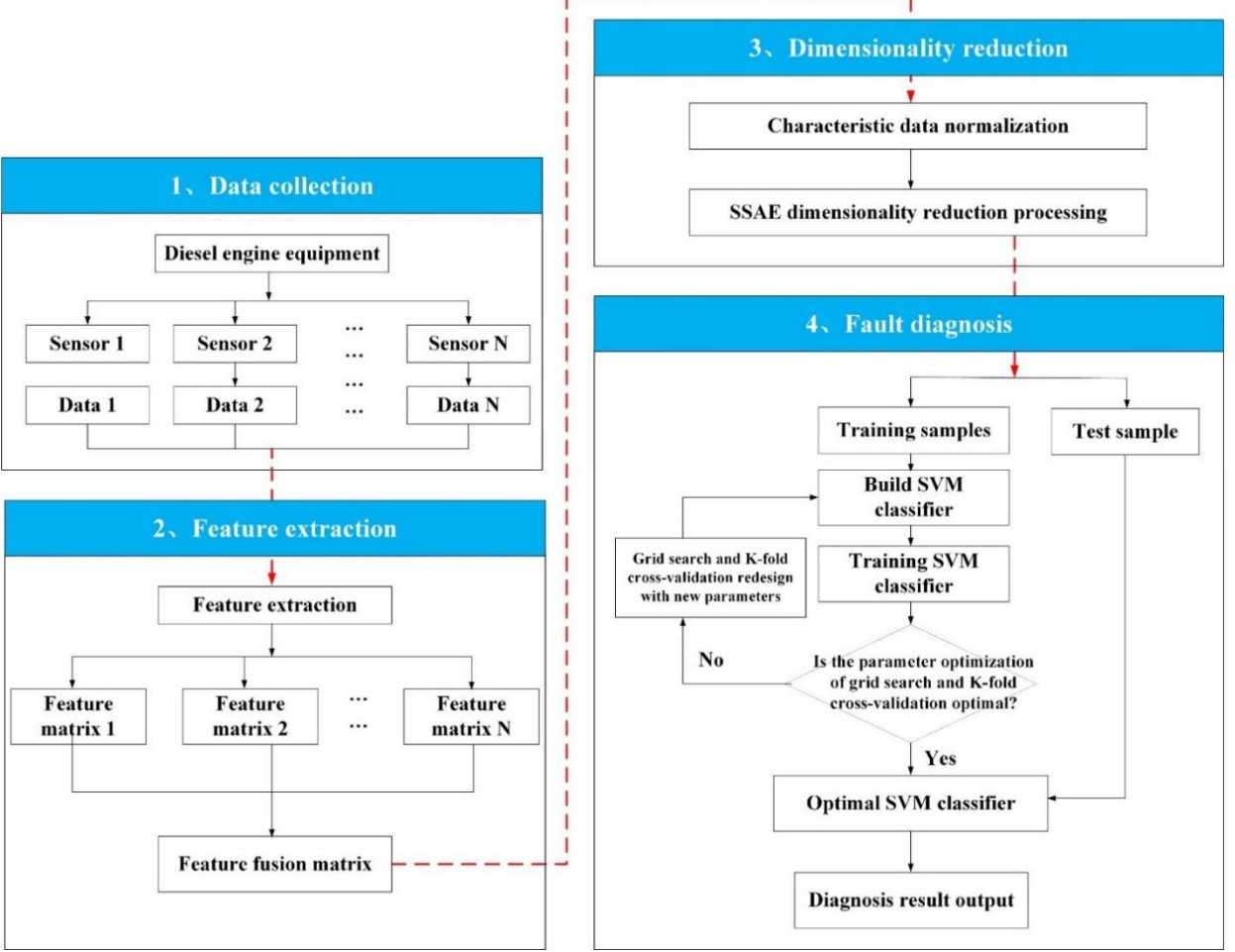

**Figure 5.** The process of the diesel engine fault diagnosis based on the SSAE-SVM method.

**Step 1: Data collection.** By using the vibration sensor, the raw vibration signal of the diesel engine was collected.

**Step 2: Feature extraction.** In the analysis of diesel engine vibration signals, the main parameters are commonly used to reflect fault characteristics, including time-domain, frequency-domain characteristics, wavelet packet energy, and the other 31 characteristic parameters, as shown in Table 1 [24–26]. Thus, the main basis for the analysis of diesel engine fault diagnosis is provided in Section 4.

**Table 1.** The parameters of the fault characteristics.

| Feature Classification | Parameter Characteristics |
|---|---|
| Time-domain features | 1 maximum value; 2 minimum; 3 peak-to-peak; 4 mean; 5 mean square; 6 roots mean square; 7 average amplitude; 8 root amplitude; 9 variances; 10 standard deviations; 11 peak; 12 kurtoses; 13 skewness; 14 energy; 15 peak indicators; 16 impulse indicators; 17 waveform indicator; 18 margin indicators; 19 clearance factor |
| Frequency domain features | 20 frequency mean; 21 frequency center; 22 RMS frequency; 23 frequency standard deviation |
| Wavelet packet energy | Wavelet packet energy features (1–8) |
| Common features | peak-to-peak; mean; mean square; variance; peak; kurtosis (6) |
| All features | Time domain feature parameters (19) Frequency domain feature parameters (4) Wavelet packet energy feature parameters (8) |

**Step 3: SSAE-based feature fusion.** The $31 \times N$ feature matrix composed of multiple sensors was utilized to eliminate the influence caused by the difference in magnitude between features. Therefore, it was necessary to perform data normalization processing on the feature matrix, to contain the data between [0, 1], so as to ensure faster convergence when the program was running. The normalized results were processed by the SSAE feature fusion. The SSAE parameter settings mainly included the number of input layer nodes, the hidden layer parameters, the weight adjustment coefficient, the sparse penalty weight, and the sparsity ratio. Extracting valuable fault feature matrices was effectively conducted to achieve the purpose of dimension reduction in the data.

**Step 4: Fault Diagnosis Based on the SVM.** First, the training samples were input into the SVM classifier. Then, the hyperparameter optimization of the SVM was carried out by utilizing a combination of grid search and K-fold cross-validation. Afterward, the optimal SVM training model was obtained. The test samples were input into the trained SVM model to verify it. Finally, the results of the fault classification were attained.

## 4. The Verification of the Experiment

### 4.1. Presetting Experimental Failure Modes

The diesel engine generates a variety of vibration signals during a cycle of reciprocating motion. The vibration signal collected by the sensor is mainly generated by the shock caused by the suction, compression, working, and exhaust strokes. In the collection of the actual vibration signals, the position of the installed sensor, noise interference, and other factors will cause problems such as data distortion or signal flooding. Therefore, a total of six vibration sensors were installed in this experiment, to facilitate data fusion analysis. The vibration frequency of diesel engines is usually in the range of 600–3000 Hz. Therefore, a new type of piezoelectric unidirectional vibration acceleration sensor was used in this study. The main parameters are provided as follows: Model BW14100; range ±50 g; sensitivity 100 mV/g; frequency range 1~8 kHz; resolution 0.0001 g. Assuming that the lower sampling frequency of the sensor was set to 1 kHz, although the vibration signal of the lower frequency could be collected normally, the research object was a diesel engine with complex structural design. In diesel engine preset fault experiments, the vibration

signal produced is non-stationary and nonlinear, which means that it usually contains a large number of strong interference signals, high frequency signals, and fault signals. However, in diesel engine preset fault experiments (for example: insufficient fuel supply of the fuel injection pump), the diesel engine will produce severe vibration and abnormal noise (these signals are all high-frequency vibration signals). During the data collection process, if a lower sampling frequency (1 kHz) is selected for data collection, it will not be possible to collect vibration signals with higher frequencies. However, the fault signals are usually contained in complex high-frequency signals, which cannot meet the conditions of subsequent feature extraction and fault diagnosis analysis. The data sampling frequency was set to 20 kHz (according to Nyquist's sampling law, the sampling frequency was set at more than twice the maximum frequency of 8 kHz in the measured signal), which could effectively collect the vibration signals of the diesel engine. A vibration sensor 1~6# was installed on the cylinder head of cylinders 1~6 along the axial direction, as shown in Figure 6. The data acquisition system consisted of an acquisition chassis, a data acquisition card (model PXI-9108, PXI-3342), and acquisition software written by Labview.

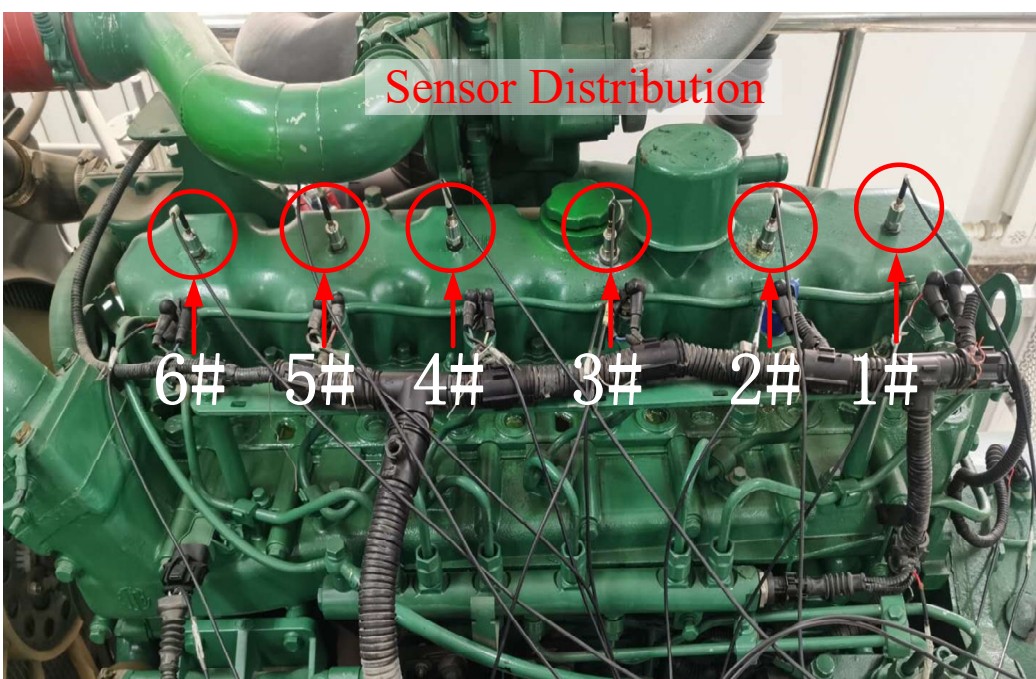

**Figure 6.** The deployment location of the vibration sensors.

Diesel engines have a complex working environment and long-term overload operations, resulting in high failure rates. To improve stability and reliability, and to reduce the failure rates of diesel engines, this paper mainly focused on the relevant research about the fuel supply system of the diesel engine. A total of six failure modes were preset, and the detailed failure mode sequence is shown in Table 2. The insufficient fuel supply of the fuel injection pump is generally caused by wear on the internal gear of the fuel injection pump, which causes the pressure of the fuel supply to decrease. In the experiment, a faulty fuel injection pump was utilized to mimic the situation. Hence, one- and six-cylinder misfires could be realized by disconnecting the injector power cord. Air filter clogging simulated clogging failure by adding an air intake cover in the air inlet. The oil supply pipe was broken, and dripping oil was realized by utilizing the faulty oil supply pipe.

**Table 2.** The preset failure modes of the diesel engine.

| Fault States | Failure Modes |
|---|---|
| 1 | Normal |
| 2 | Insufficient fuel supply from the fuel injection pump |
| 3 | One-cylinder misfire |
| 4 | Six-cylinder misfire |
| 5 | Air filter was clogged |
| 6 | The oil supply pipe was broken, with dripping oil |

*4.2. Presetting the Description of the Fault Test Data*

The ignition sequence of each cylinder of the diesel engine was 1-5-3-6-2-4. In the experiment, the data collection was started just after the diesel engine was warmed up, to ensure stable data collection, and the rotational speed of the diesel engine was uniformly set to 800 rpm. In accordance with the Nyquist sampling law, the sampling frequency of the data acquisition was set to more than twice the maximum frequency of 8 kHz in the measured signal—that is, 20 kHz. The duration of each collection was set to 12 s, which was recorded as one group of data, so that 30 groups of data were collected for each failure mode. Then, collection of a set of data every 30 s interval was set. The details are shown in Table 3.

**Table 3.** The pre-built six types of fault data sets.

| Fault States | Rotating Speeds | Number of Sensors | Sampling Frequency | Sampling Time | Number of Samples |
|---|---|---|---|---|---|
| 1 | 800 rpm | 6 | 20 kHz | 12 s | 30 |
| 2 | 800 rpm | 6 | 20 kHz | 12 s | 30 |
| 3 | 800 rpm | 6 | 20 kHz | 12 s | 30 |
| 4 | 800 rpm | 6 | 20 kHz | 12 s | 30 |
| 5 | 800 rpm | 6 | 20 kHz | 12 s | 30 |
| 6 | 800 rpm | 6 | 20 kHz | 12 s | 30 |

*4.3. The Comparative Analysis of the Parameter Characteristics*

The fault parameter characteristics presented in Section 3 were adopted, and the parameter characteristics were divided into five types: common features (6); time-domain features (19); frequency domain features (4); wavelet energy features (8); and all features (31). By utilizing the data set in Section 4.2, there existed 30 groups of samples. Each group of samples had 240,000 sampling points. Then, 20,000 sampling points were taken as a group, so that $12 \times 30 = 360$ data samples could be formed. The six fault states led to a total of $6 \times 360 = 2160$ samples. The six fault states referred to the feature matrix extracted from the ignition sequence 1-5-3-6-2-4 of the diesel engine, as shown in Table 4. To verify the effectiveness of these feature parameters, five types of feature parameters were utilized as input variables. The Matlab2020 software development tool was utilized to verify the SSAE-SVM method.

**Table 4.** The data set of the fault parameter characteristics.

| Serial Number | Feature Taxonomy Combination | Characteristic Parameters | Feature Matrix | Number of Sensors |
|---|---|---|---|---|
| 1 | Common features | 6 | $6 \times 2160$ | 6 |
| 2 | Time domain features | 19 | $19 \times 2160$ | 6 |
| 3 | Frequency domain features | 4 | $4 \times 2160$ | 6 |
| 4 | Time domain + frequency domain features | 23 | $23 \times 2160$ | 6 |
| 5 | Wavelet packet energy | 8 | $8 \times 2160$ | 6 |
| 6 | Common features + frequency domain features | 10 | $10 \times 2160$ | 6 |
| 7 | Common features + wavelet packet energy | 14 | $14 \times 2160$ | 6 |
| 8 | All features | 31 | $31 \times 2160$ | 6 |

Both Figure 7 and Table 5 show that the penalty factor c and the kernel parameter g utilized a combination of grid search and K-fold cross-validation. Then, both optimal hyperparameters c and g were resolved; where the parameter c varied in [−10, 10], the parameter g altered in [−10, 10], and the search step size was uniformly set to 0.1. Only the accuracy rate of Combination 5 reached 77%, and the rest were found to be more than 90%, indicating that the parameter characteristics of Combination 5 were not suitable for fault diagnosis among the eight combinations. Furthermore, the cross-validation accuracy and fault classification accuracy of Combinations 6 and 7 were found to be above 99%. When compared with Combination 7, Combination 6 had the advantages of shorter optimization time and fewer parameter characteristics. However, the cross-validation and fault classification accuracy of Combination 2 reached over 98%. When compared with Combinations 6 and 7, the search time took longer, and there were more parameter characteristics. Furthermore, Combination 3 had the fewest parameter characteristics and the shortest optimization time, but the accuracy of the fault classification was not as high as that of Combination 6. Combination 8 had 31 parameter characteristics, which took the longest optimization time, and the fault accuracy rate was lower than that of Combination 6, indicating that some parameter characteristics did not contribute much to the accuracy of the fault classification.

When a comprehensive analysis was conducted, not all feature parameters were found to be effective, and appropriate features had to be selected for fault diagnosis to reduce the computational resources needed, and to improve classification accuracy. When compared with other combinations, Combination 6 had the advantages of fewer parameter characteristics, a short optimization time, and higher precision. Thus, it was fully demonstrated that Combination 6 contained more useful signals, and was very suitable to utilize in the fault diagnosis of a diesel engine.

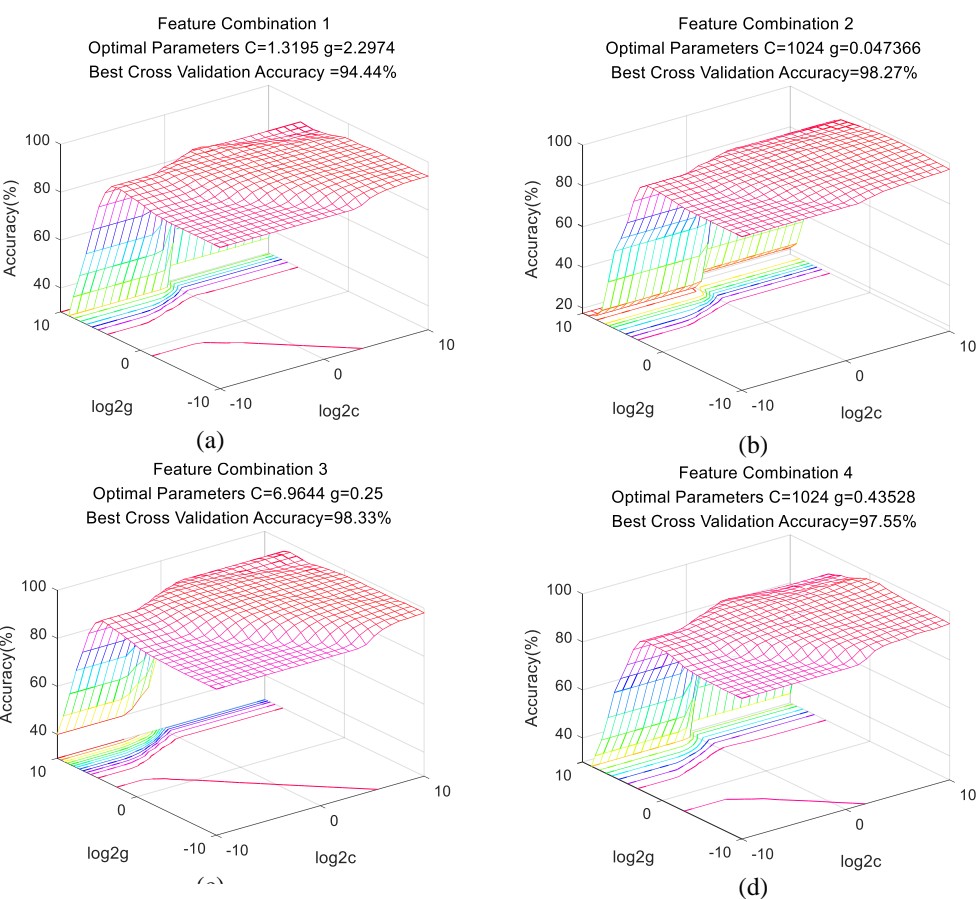

**Figure 7.** *Cont.*

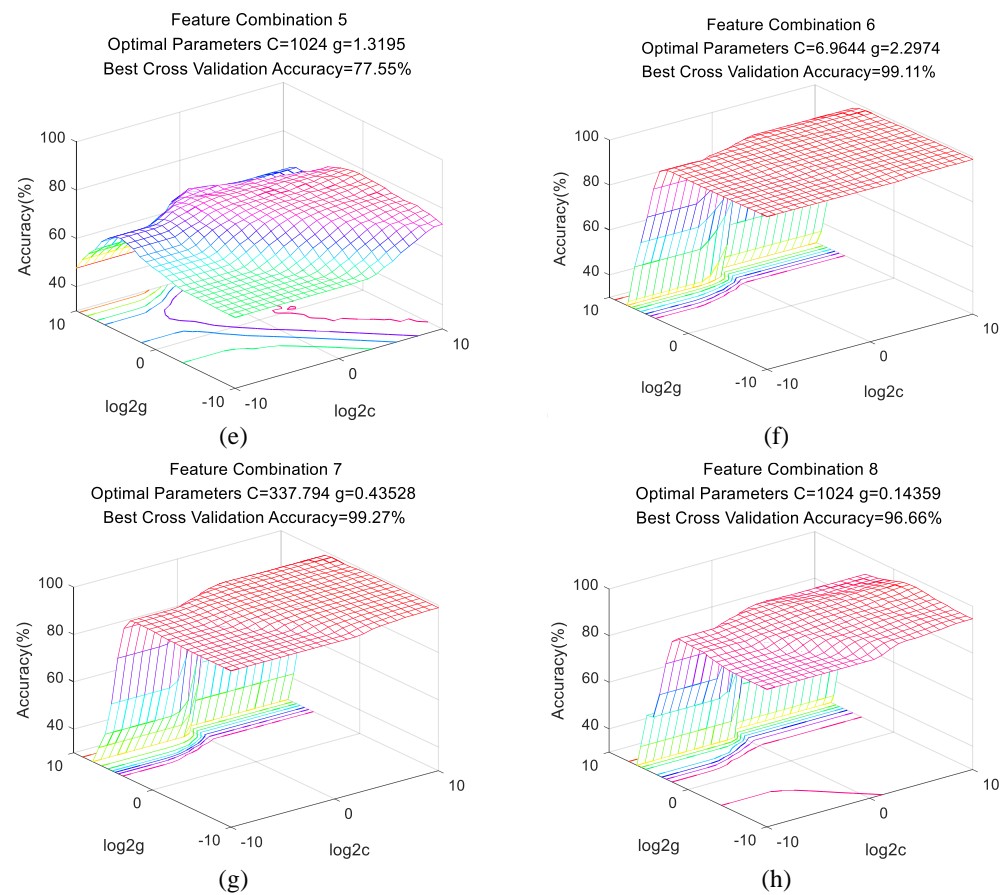

**Figure 7.** The best optimization results of parameters *C* and *g* under different parameter characteristics: (**a**) Feature Combination 1; (**b**) Feature Combination 2; (**c**) Feature Combination 3; (**d**) Feature Combination 4; (**e**) Feature Combination 5; (**f**) Feature Combination 6; (**g**) Feature Combination 7; (**h**) Feature Combination 8.

**Table 5.** The results of the SSAE-SVM classification when different parameter features were used.

| Feature Taxonomy Combinations | Number of Features | After Optimization ($C, g$ Values) | Cross Validation Accuracy | Diagnostic Accuracy | Execution Time |
|---|---|---|---|---|---|
| Combination 1 | 6 | (1.3195, 2.2974) | 94.44% | 94.72% | 201.2542 s |
| Combination 2 | 19 | (1024, 0.047366) | 98.27% | 99.16% | 403.2165 s |
| Combination 3 | 4 | (6.9644, 0.25) | 98.33% | 97.50% | 192.6870 s |
| Combination 4 | 23 | (1024, 0.43528) | 97.55% | 96.38% | 400.4194 s |
| Combination 5 | 8 | (1024, 1.3195) | 77.55% | 77.22% | 274.2213 s |
| Combination 6 | 10 | (6.9644, 2.2974) | 99.11% | 99.44% | 205.4491 s |
| Combination 7 | 14 | (337.794, 0.43528) | 99.27% | 99.44% | 297.8969 s |
| Combination 8 | 31 | (1024, 0.14359) | 96.66% | 96.11% | 561.9832 s |

### 4.4. The Comparative Analysis of the Dimension Reduction Methods

To prove the effectiveness of the SSAE in the extraction of the fusion features by utilizing the SSAE-SVM method, the sensors 1-5-3-6-2-4# data were selected, and Combination 6 was employed, to form a $60 \times 2160$ feature matrix to conduct verification according to the ignition sequence of each cylinder of the diesel engine. After fusion by the SSAE method, the number of SSAE input nodes was set to 60; the hidden layer parameters were set to 30 and 10, respectively; the sparsity ratio was set to 0.1; the weight adjustment coefficient was set to 0.000002; and the sparse penalty weight was set to 0.0002. Thus, a new $10 \times 2160$

feature matrix was attained, and it was divided into 1800 training samples and 360 testing samples, respectively.

To further prove the effectiveness of the SSAE for extracting fusion features, the data set was also selected from the sensors 1-5-3-6-2-4# data, and the feature parameters were combined with 6 to form a $60 \times 2160$ feature matrix to conduct verification. The SSAE, PCA, and KPCA methods were utilized to conduct the dimension reduction of the features for the comparative analysis. In the Matlab2020 software development tool, a combination of both grid search and K-fold cross-validation was employed. Hyperparameter optimization on penalty factor $c$ and kernel parameter $g$ was performed, where parameter $c$ took values in $[-10, 10]$, and parameter $g$ changed in $[-10, 10]$. Then, the three feature fusion matrices were imported into the SVM model for training. Thereby, three optimal training models were obtained. Finally, the test samples were plugged into the trained model for fault identification, and the fault diagnosis accuracy results were obtained eventually, as shown in Table 6. Figures 8–10 and Table 6 depict that the diagnostic accuracy rates of the SSAE, the PCA, and the KPCA reached 99.44%, 84.16%, and 94.44%, respectively. The SSAE method had the highest diagnostic accuracy and the shortest search time. However, when compared with the other two methods, the cross-validation accuracy and the diagnostic accuracy of the PCA method were significantly lower. Therefore, the results show that the SSAE-SVM method can effectively extract valuable fusion feature parameters, and has better advantages and generalizability in fault diagnosis and identification.

**Table 6.** The classification results of different dimension reduction methods.

| Dimensionality Reduction Method | After Optimization ($C$, $g$ Values) | Cross Validation Accuracy | Diagnostic Accuracy | Execution Time |
|---|---|---|---|---|
| SSAE-SVM | (6.9644, 2.2974) | 99.11% | 99.44% | 205.4491 s |
| PCA-SVM | (1024, 4) | 85.66% | 84.16% | 232.9893 s |
| KPCA-SVM | (4, 6.9644) | 94.50% | 94.44% | 219.0951 s |

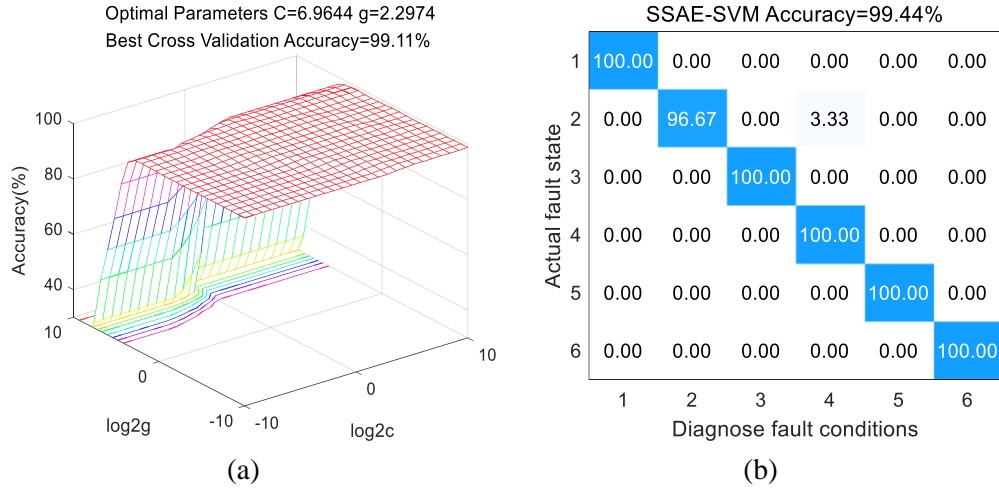

(a)          (b)

**Figure 8.** The SSAE dimension reduction method: (**a**) optimal parameters; (**b**) classification results.

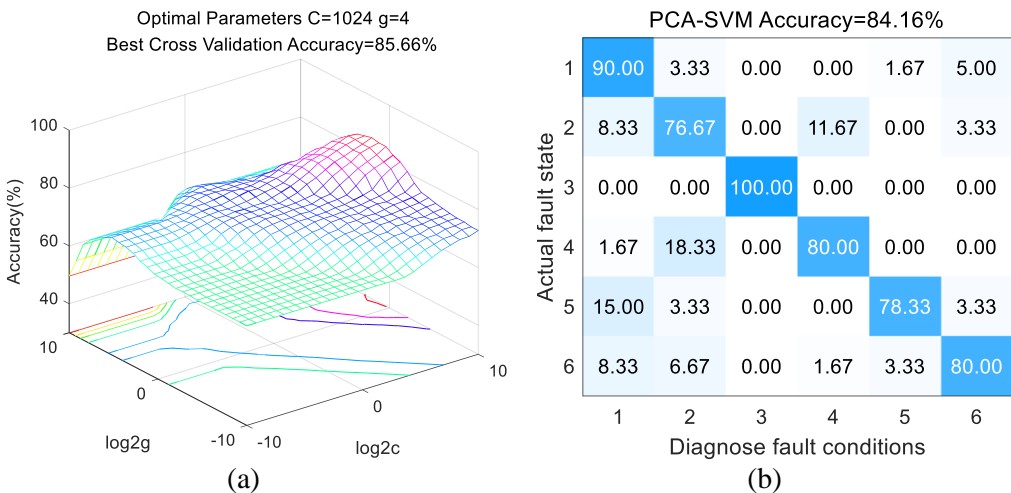

**Figure 9.** The PCA dimension reduction method: (**a**) optimal parameters; (**b**) classification results.

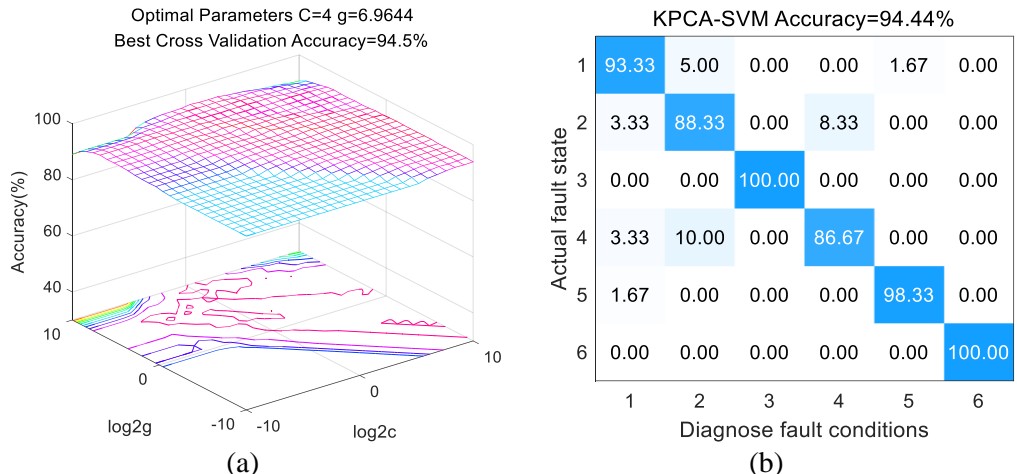

**Figure 10.** The KPCA dimension reduction method: (**a**) optimal parameters; (**b**) classification results.

### 4.5. The Comparative Analysis of the Sensor Combination

To prove the effectiveness of the SSAE-SVM fault diagnosis method, the dataset of the feature parameters of Combination 6 (common features + frequency domain features) in Section 4.3 was utilized. Because there exist many combination methods of sensors, this paper presents just a few of them, due to the limited scope of the research. Therefore, this subsection assumes that the sensor combination was established based on the ignition sequence 1-5-3-6-2-4 of each cylinder of the diesel engine, and six different sensor combinations were set to conduct verification. After the fusion of the SSAE method, the main parameter settings of the SSAE were the same as those of the SSAE in Section 4.4. A new $10 \times 2160$ feature matrix was obtained, which was divided into 1800 training samples and 360 test samples, respectively, as shown in Table 7.

Figure 11 and Table 8 depict that the cross-validation accuracy and diagnostic accuracy of Combination 6 were 77.55% and 76.38%, respectively. It shows that the effect of employing sensor data for fault diagnosis was very poor, and could not meet the requirements of fault diagnosis. When compared with Combination 6, the diagnostic accuracy of Combination 5 was greatly improved, and the accuracy rate reached 96.94%, indicating that the utilization of the two sensor data contained the key information for diagnosis. The fault diagnoses for Combinations 1 to 4 were all above 99% accurate, indicating that the more data samples are collected, the higher the fault diagnosis accuracy will be. When compared with the other three combinations, Combination 3 had the advantages of short optimization time, higher diagnostic accuracy, and a small number of sensors. Although Combination 4

had one less sensor than Combination 3, the diagnostic accuracy also reached 99.44%. By considering the complex working conditions and nonlinear factors of the diesel engine, the higher diagnostic accuracy of Combinations 1 and 2 was attained. However, due to the large number of sensors and the high cost, it was not suitable for fault diagnosis. Due to the small number of sensors in Combinations 5 and 6, the diagnostic accuracy was low, and so it was not suitable for diesel engine fault diagnosis. Combination 3 had higher diagnostic accuracy, fewer sensors, and lower cost, and the installation position was reasonable (deployed according to the ignition sequence of each cylinder of the diesel engine). Therefore, Combination 3 was more suitable for fault diagnosis. The results suggest that employing the SSAE-SVM method to diagnose the data characteristics related to Combination 3 had the advantages of short optimization time, better effect, and lower cost.

**Table 7.** The data sets with different sensor combinations.

| Combination Numbers | Combinations | Number of Sensors | Feature Matrices | Input Nodes | Hidden Layer Parameters |
|---|---|---|---|---|---|
| Combination 1 | 1-5-3-6-2-4# | 6 | 60 × 2160 | 60 | (30, 10) |
| Combination 2 | 1-5-3-6-2# | 5 | 50 × 2160 | 50 | (20, 10) |
| Combination 3 | 1-5-3-6# | 4 | 40 × 2160 | 40 | (20, 10) |
| Combination 4 | 1-5-3# | 3 | 30 × 2160 | 30 | (20, 10) |
| Combination 5 | 1-5# | 2 | 20 × 2160 | 20 | (15, 10) |
| Combination 6 | 1# | 1 | 10 × 2160 | 10 | (10, 10) |

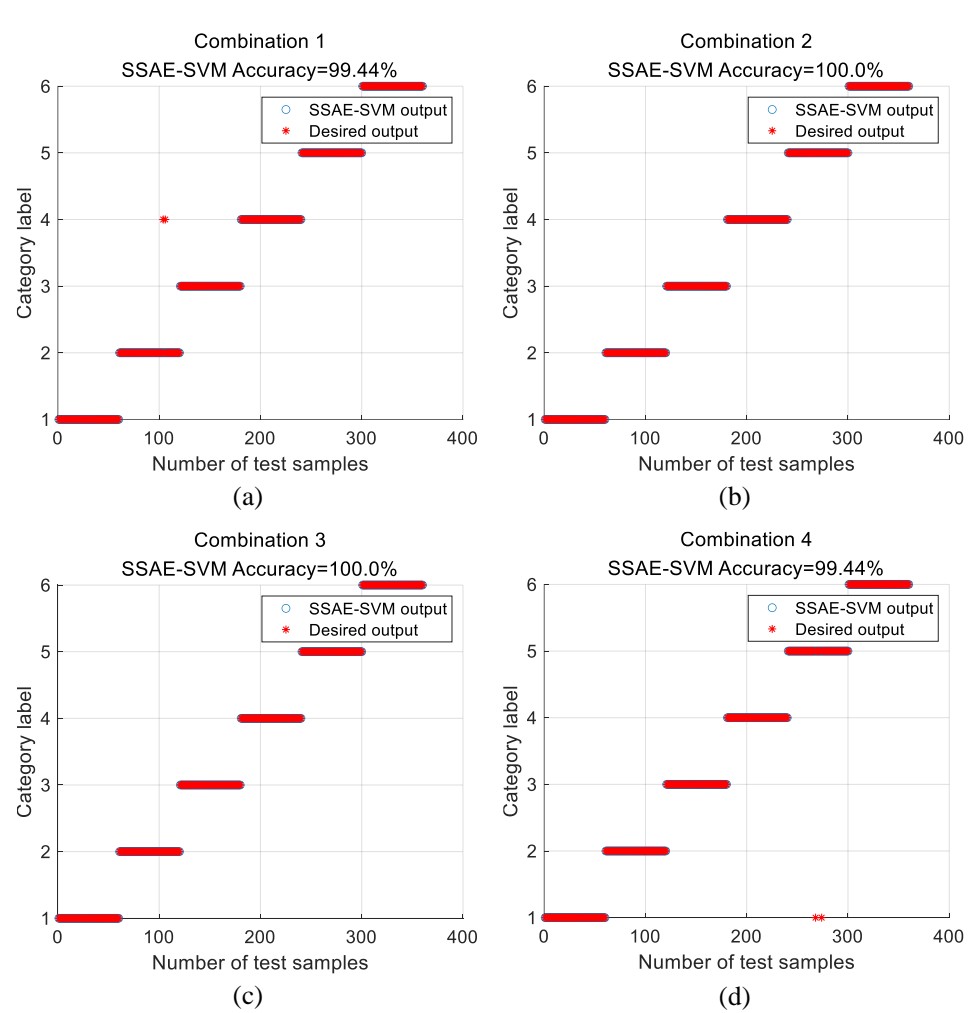

**Figure 11.** *Cont.*

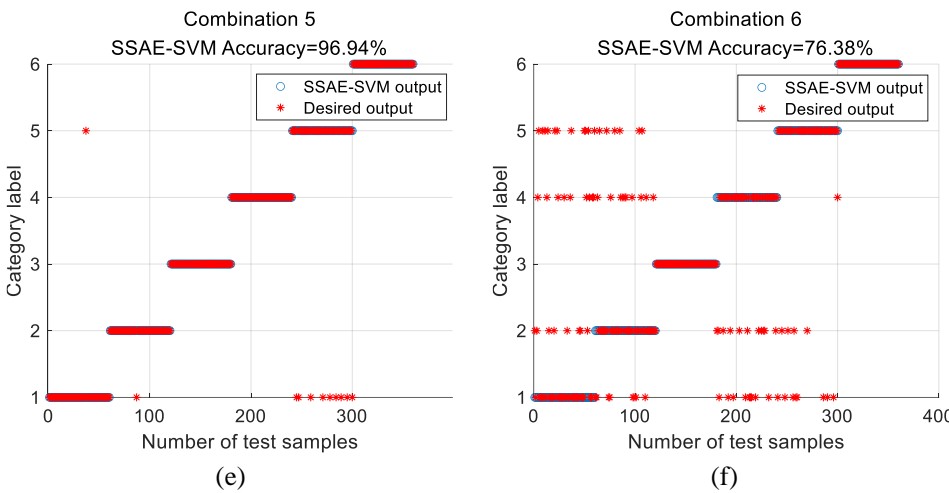

**Figure 11.** The SSAE-SVM diagnosis results under different sensor combinations: (**a**) Combination 1; (**b**) Combination 2; (**c**) Combination 3; (**d**) Combination 4; (**e**) Combination 5; (**f**) Combination 6.

**Table 8.** The comparative analysis of the diagnostic results under different sensor combinations.

| Combination Numbers | Combinations | After optimization ($C$, $g$ Values) | Cross Validation Accuracy | Diagnostic Accuracy | Execution Time |
|---|---|---|---|---|---|
| Combination 1 | 1-5-3-6-2-4# | (6.9644, 2.2974) | 99.11% | 99.44% | 255.4491 s |
| Combination 2 | 1-5-3-6-2# | (0.435275, 0.75786) | 99.94% | 100.0% | 260.6388 s |
| Combination 3 | 1-5-3-6# | (6.9644, 2.2974) | 99.88% | 100.0% | 208.4682 s |
| Combination 4 | 1-5-3# | (1.31951, 4) | 99.94% | 99.44% | 252.1299 s |
| Combination 5 | 1-5# | (1024, 0.43528) | 97.61% | 96.94% | 248.1764 s |
| Combination 6 | 1# | (12.1257, 2.2974) | 77.55% | 76.38% | 288.8677 s |

### 4.6. The Comparative Analysis of the Classification Methods

To further verify the effectiveness of the SSAE-SVM fault diagnosis method proposed in the manuscript, the data set of feature parameter Combination 6 (common features + frequency domain features) in Section 4.3 was utilized and chosen to form a 40 × 2160 feature matrix for verification, according to Combination 3, the optimal combination of sensors in the previous section. After the fusion of the SSAE method, the main parameter settings of the SSAE were the same as those of the SSAE parameter settings in Section 4.4. Therefore, a new 10 × 2160 feature matrix was obtained, which was divided into 1800 training samples and 360 test samples, respectively. The SVM, Decision Tree (DT), Naive Bayes Classifier (NBC), and Random Forest (RF) methods were employed to diagnose faults on the training samples. and were compared and analyzed for test samples. The results are presented in Table 9.

**Table 9.** The comparative analysis of the diagnostic accuracy of different classification methods.

| Fault States | SVM | DT | NBC | RF |
|---|---|---|---|---|
| 1 | 100.0% | 93.33% | 90.00% | 96.67% |
| 2 | 100.0% | 95.00% | 93.33% | 98.33% |
| 3 | 100.0% | 100.0% | 100.0% | 100.0% |
| 4 | 100.0% | 96.67% | 100.0% | 100.0% |
| 5 | 100.0% | 88.33% | 95.00% | 98.33% |
| 6 | 100.0% | 100.0% | 100.0% | 100.0% |
| Accuracy | 100.0% | 95.55% | 96.38% | 98.88% |
| Execution Time | 208.4682 s | 223.6581 s | 252.2783 s | 237.1265 s |

Both Figure 12 and Table 9 show that the accuracy rates of the three classification methods of DT, NBC, and RF were: 95.55%, 96.38%, and 98.88%, respectively. The SSAE-SVM fault diagnosis method proposed in this study had an accuracy rate of 100%, and the time spent was shorter than those of the other three classification methods. The DT, NBC, and RF methods had a lower ability to identify the normal state, fault states 1 and 4, indicating that those three methods were insufficient to diagnose when a stronger noise environment existed. The results show that the SSAE-SVM fault diagnosis method has higher accuracy when compared with other methods, Therefore, the validity of the SSAE-SVM fault diagnosis method is presented, and the theoretical basis for diesel engine fault diagnosis is provided.

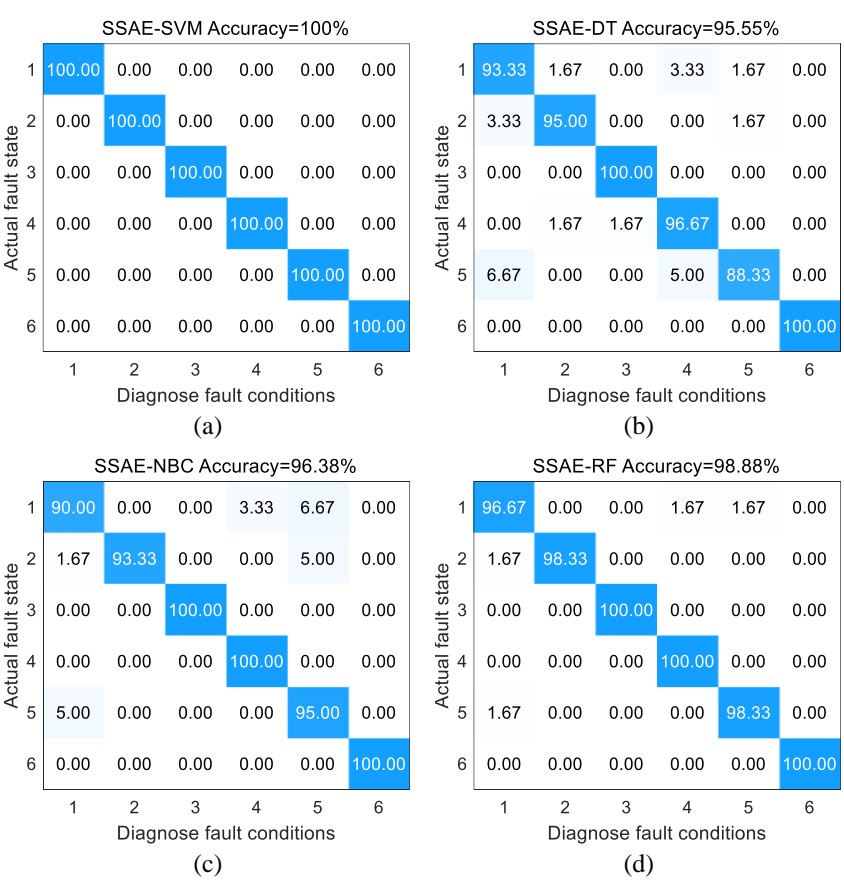

**Figure 12.** The diagnosis results of the different classification methods: (**a**) SVM = 100.0%; (**b**) DT = 95.55%; (**c**) NBC = 96.38%; (**d**) RF = 98.88%.

## 5. Conclusions

This paper proposes a fault diagnosis method based on the SSAE-SVM, which can be applied to the fault diagnosis of diesel engines in complex environments. The SSAE method was utilized to effectively improve the feature extraction ability of nonlinear data, to achieve the purpose of feature dimension reduction. By utilizing both grid search and the K-fold cross-validation method to optimize the hyperparameters of the SVM method, the fault classification effect was effectively improved.

The experimental results show that the SSAE-SVM method proposed in this research can effectively obtain a higher diagnostic rate when employing both fewer sensors and eigenvalues. Thus, this method not only reduces the cost of fault diagnosis but also provides a reference for the optimal arrangement of sensors.

**Author Contributions:** Data curation, H.B. and X.Z.; Resources, L.W. and H.Y.; Supervision, X.J.; Validation, L.W. and X.J.; Writing—original draft, H.B.; Writing—review & editing, H.B. and Y.Y.; All authors have read and agreed to the published version of the manuscript.

**Funding:** This study did not receive any funding.

**Institutional Review Board Statement:** Not applicable.

**Informed Consent Statement:** Not applicable.

**Data Availability Statement:** Data will be provided upon request to the authors.

**Conflicts of Interest:** Authors declare that they have no conflict of interest.

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
