# Peer review of "Research on Diesel Engine Fault Diagnosis Method Based on Stacked Sparse Autoencoder and Support Vector Machine"

_electronics, doi:10.3390/electronics11142249_

Round 1

Reviewer 1 Report

Dear Authors,

The paper focuses on the use of deep learning using coders and SVMs to determine the degradation of a device, in this case, an internal combustion engine. As such, the article is part of a trend that is currently evident.

However, the way the assumptions for the analyses and results are presented is far from acceptable.

The description presented in section 2 is known and available in the literature. There is no information why such and not other parameters determining the engine condition were chosen and to which groups they were classified. The authors provide some classification in Table 4, however, there is no clear reference to the parameters previously declared for analysis (Table 1). Additionally, it should be noted that the article was submitted to the Electronics journal and the primary object of study is the engine. Therefore, more attention should have been paid to clarifying mechanical (engine) terms.

It should be noted that the authors tried to approach the issue of the influence of individual groups of parameters on the analysis results, however in a completely illegible and for me incomprehensible way, e.g. Fig. 7 Combination 1 - 8, Fig. 11 Combination 1-6 - these are not the same sets of parameters and their presentation is illegible. If one analyses signals from e.g. 2 sensors corresponding to pistons 1 and 5 having 6 sensors, it would have to be demonstrated that this is an appropriate choice of sensors. Which has not been done.

The results presentation section should be thoroughly rebuilt.

Much of the paper is based on determining the optimal parameters c and g that play an important role in the classification case. However, a more detailed description of these parameters is lacking.

The use of a vibration sensor with a lower range of 1 kHz is also puzzling, despite the fact that the authors themselves point out the vibration range occurring in the engine from 600 Hz. In this range, there is no proper comment or it would be necessary to repeat the experiments with a sensor of a proper measuring range. Apart from that, it would be more appropriate to investigate the influence of injector spoiling on the obtained results than its complete disconnection. This would allow for timely prevention.

Moreover:

1. In many cases the information given in the text and tables overlap (e.g., l. 244- 249 and Table 3).

2. Figures 5 left and 6 do not contribute significant content to the paper and should be removed.

3. From Figure 5 right, especially in the paper, it is difficult to determine the location of the sensors.

4. There is no information on what software was used in the analyses conducted.

5. Figures 7, 8, 9, and 10 are poorly colored, which results in poor legibility of the figures

6. There are numerous stylistic and language errors in the article

Author Response

The responses are attached in the file

Reviewer 2 Report

The authors propose an diesel engine fault diagnosis method based on Stacked Sparse Autocoder and Support Vector Machine. It was shown that the number of sensor is a key feature to achieve high diagnosis accuracy and obtained a high diagnosis accuracy with 3 to 6 sensors. Also reducing the cost of diagnosing engine fault. The manuscript is well written and present enough new results in order to deserve publication.

I advise the authors to comment on the choice of number and position of sensors. Is there a evaluation of how the position of the sensors change accuracy?

In conclusion, the manuscript has my recommendation in order to be published in Electronics.

Author Response

The responses are given in the file. 

Round 2

Reviewer 1 Report

Dear Authors,

you have written:

“The vibration frequency of diesel engines is usually in the range of 600-3000 Hz. Therefore, a new type of piezoelectric unidirectional vibration acceleration sensor is used in this paper. The main parameters are provided as follows: Model BW14100, range ±50g, sensitivity 100 mV/g, frequency range 1~8kHz, resolution 0.0001g. The data sampling frequency is set to 20kHz (According to Nyquist’s sampling law, the sampling frequency is set at more than twice the maximum frequency of 8kHz in the measured signal), which can effectively collect vibration signals of the diesel engine.”

 However, there is still no answer to the question of whether a sensor with a lower limit of 1kHz can be used? Please respond to this question.

 I have marked the other comments in the text.

Author Response

Please find the attached response letter

Round 3

Reviewer 1 Report

Dear Authors,

You have not yet taken into account the editing comments I sent previously:

1. sometimes the parameter "C" is described by "c", mainly in the figures

2. in the description of the literature there is often no space before "[J]"

3. l. 470 two dots

4. l.472 space before ":"

5. l. 457 and 453 space before "."
